Global species richness of dragonflies and damselflies (Odonata): latitudinal trends and insular colonization

Cordero-Rivera Adolfo adolfo.cordero@uvigo.es
Departamento de Ecoloxía e Bioloxía Animal, University of Vigo , Pontevedra , Galiza , Spain
Erasmus Daniel
Electronic publication date: 2025 Sep 15
Publication date: 2025
Volume: 13
Electronic Location ID: e20004
Received 2025 May 5; Accepted 2025 Aug 6
Copyright: ©2025 Cordero-Rivera
Copyright year: 2025
Copyright holder: Cordero-Rivera
License: This is an open access article distributed under the terms of the Creative Commons Attribution License, which permits unrestricted use, distribution, reproduction and adaptation in any medium and for any purpose provided that it is properly attributed. For attribution, the original author(s), title, publication source (PeerJ) and either DOI or URL of the article must be cited.
License URL: https://creativecommons.org/licenses/by/4.0/

Keywords: Biogeography, Aquatic insects, Islands, Area effects, Distance to continent, Elevation

Funding: MICIU/AEI/10.13039/501100011033 PID2023-147268NB-I00 ERDF A way of making Europe This work was funded by grant PID2023-147268NB-I00 from MICIU/AEI/10.13039/ 501100011033 and by “ERDF A way of making Europe”. The funders had no role in study design, data collection and analysis, decision to publish, or preparation of the manuscript.

==============================
Odonates are mainly “sun lover” insects, and therefore reach maximum diversity in tropical regions, particularly in rainforests. Here, worldwide patterns of species richness of the order are analysed, by compiling a database of species numbers for 255 continental regions and 243 islands. Area, distance to continents and elevation for all islands were estimated, and their effect analysed on odonate species richness by means of linear models. As expected, a clear effect of latitude and insularity on the species richness of Odonates was found, with a maximum of  550 species in the equator for continental areas but only  200 species in islands. In islands, latitude, area and distance to the continent clearly affect species richness, but elevation had no significant effect. The continental countries with highest richness are Venezuela (548) and Colombia (543 species). Brazil (863) and China (818) have higher richness, but given their size were included in the analysis as states and regions. Excluding very large islands (New Guinea, Sumatra or Borneo), which are considered continents in this paper, Japan (209) is the archipelago with highest richness, albeit Indonesia (737) and the Philippines (306) have more species, but were analysed subdivided by islands. The proportion of Zygoptera found at the different regions was negatively affected by latitude and positively by the area, but not by insularity. In contrast, in islands the proportion of Zygoptera was not affected by latitude, distance or elevation, but was positively affected by area. These analyses highlight the ability of odonates to colonize even the most remote islands, places that can be sources of rapid speciation, as occurred in Hawaii or Fiji.

Introduction

The latitudinal gradient of species richness, i.e., the fact that more species are found per unit area near the Equator and their number gradually diminishes towards the poles (reviewed by Willig, Kaufman & Stevens, 2003), is among the best established patterns in Ecology and can clearly be considered a general law (in the sense of Colyvan & Ginzburg, 2003). The pattern is known to occur in all ecosystems, but is stronger in terrestrial and marine ecosystems than in freshwater environments, and more pronounced in larger organisms (Hillebrand, 2004). The causes behind this pattern are however still debated (Schemske & Mittelbach, 2017), almost 60 years after the publication of Pianka’s (1966) seminal paper that compiled and reviewed the six main hypotheses that are still at the centre of the debate (Rohde, 1992).

In the case of insects, the general pattern is that their maximum diversity is found in tropical areas (Willig, Kaufman & Stevens, 2003) but there are some examples of groups that do not exhibit the expected decline in diversity with increasing latitude or even show opposite patterns (Gaston, 1996; Kindlmann, Schödelbauerová & Dixon, 2007; Heino, 2011). Considering herbivore insects, the expectation is that their diversity will follow that of plants, which clearly increases at lower latitudes (Terborgh, 1992). In fact the effect of herbivory on plant biomass is stronger in the tropics (Schemske et al., 2009). Predacious insects will therefore also have more resources to feed on in tropical ecosystems, resulting in more diverse biotic interactions at low latitudes (Schemske et al., 2009).

Freshwater ecosystems have received less attention than terrestrial ones in relation to the latitudinal gradient of biodiversity, and latitudinal patterns of odonate species richness at a global scale have not been studied in detail. Only 23 years ago, a short paper noted that Central America had 29 times more species of Odonata and 13 times more Ephemeroptera per unit area than North America (Boyero, 2002). At that time, our knowledge of Odonata diversity in tropical America was still very fragmentary, but that “rough” comparison was among the first to clearly highlight that some aquatic insects followed the expected latitudinal pattern in species richness. Although at a regional scale strong latitudinal patterns in species richness have been found for stream insects (Ephemeroptera, Plecoptera and Trichoptera) in Europe (Shah et al., 2015) and for Ephemeroptera in New Zealand (Pohe, Winterbourn & Harding, 2024), at a global scale, these gradients were found to vary markedly among taxa, with higher diversity at higher latitudes for Ephemeroptera and Plecoptera, no significant gradients for Trichoptera but very strong gradients for Odonata, with the highest diversity at low latitudes (Pearson & Boyero, 2009; Heino, 2011).

The Odonata (Fig. 1) were the first insect order to be evaluated using the International Union for Conservation of Nature (IUCN) criteria, revealing that 10% of species are at risk of extinction, which is rather high for an order “with above-average dispersal ability and relatively wide distribution ranges” (Clausnitzer et al., 2009). Their global diversity has been assessed, with the identification of areas of higher species richness, hot-spots for endemism and conservation issues (Kalkman et al., 2008). Recently, the availability of country check-lists and regional faunistic guides for Odonata has exploded, with several field guides available for most European (e.g., Grand & Boudot, 2006; Maravalhas & Soares, 2013; Riservato et al., 2014) and North American countries and states (e.g., Paulson, 2009; Paulson, 2011), comprehensive identification guides for large regions of Africa (e.g., Samways, 2008; Dijkstra & Clausnitzer, 2014), Asia (e.g., Hämäläinen & Pinratana, 1999; Orr & Hämäläinen, 2007; Subramanian, 2009; Zhang, 2019; Boudot et al., 2021), America (Lencioni, 2005; Lencioni, 2006; Garrison, Von Ellenrieder & Louton, 2006; Garrison, Von Ellenrieder & Louton, 2010; Von Ellenrieder & Garrison, 2007), Oceania (Theischinger & Hawking, 2006; Marinov & Ashbee, 2019) and some species-rich islands (Meurgey & Picard, 2011; Ozono, Kawashima & Futahashi, 2012; Reels & Zhang, 2015; Dijkstra & Cohen, 2021). However, most Neotropical diversity of Odonata still remains largely unknown, as only 1% of one-degree cells have been adequately sampled (Alves-Martins et al., 2024). This lack of knowledge precludes detailed biogeographical analysis, but macro-ecological patterns can be addressed with some confidence. This paper compiles and analyses a database of the number of species of Odonata for continental regions and islands, with the aim to test (1) the effect of latitude on Odonata species richness, (2) the effect of insularity and (3) the main predictions of the Theory of Island Biogeography for odonate communities, related to island area, distance to the continent and elevation.

Figure 1 Damselflies (Zygoptera; A, B) and Dragonflies (Anisoptera; C, D) have clear morphological differences, which are thought to predict their dispersal ability.

However, in both suborders there are species able to colonize oceanic islands, like parthenogenetic Ischnura hastata from the Azores (A) or the only cosmopolitan dragonfly, Pantala flavescens, here photographed in Cuba (C), and species restricted to primary forests, like Denticulobasis garrisoni (B) or Libellula herculea (D) both from Tiputini Biodiversity Station, Ecuador. Photos by the author.

Materials & Methods

A database of the number of species per country and/or region was constructed by an exhaustive search in faunistic and taxonomic books about Odonata, complemented by a search on Google Scholar using the keywords “checklist” or “list” and “odonat*” in April 2015. This database, originally designed to analyse the distribution of species of the genus Ischnura (see Cordero-Rivera, 2015), was updated in March 2025, repeating the same searches in Google Scholar (until page 15) and consulting taxonomic monographs and atlases published since 2015. The database was completed by a search for “island” and “odonat*” to compile species numbers for as many islands as possible. Sandall, Pinkert & Jetz (2022) provide a database of literature records and occurrence data from different sources, including lists of species per country. They also provide a list of species inferred to occur in a country when recorded from neighbouring countries, but these species are not considered here, as such an automatic assignation is likely to incur in many errors.

Given that the goal was to explore latitudinal patterns and island biogeography, in many cases a country list is not a good representation of the fauna for these purposes, because island territories or regions politically belonging to countries situated in other continents produced inflated species lists, and pooled fauna from different biogeographical regions. The present list includes islands from where an exhaustive checklist was available, irrespective of their political status as an independent nation. Therefore, a comparison of the species count of the above mentioned database was made with the species list of Sandall, Pinkert & Jetz (2022) and counts were updated when possible. In case of clear discrepancy on the number of species, priority was given to recent species lists published by odonatologists. When the lists were very different, a check of the list of Sandall, Pinkert & Jetz (2022) was made to try to detect erroneous data, like species wrongly identified (see Vega-Sánchez et al., 2020), not an uncommon fact in databases like Global Biodiversity Information Facility (GIBF) (Zizka et al., 2020), which was the main repository used for occurrence data in Sandall, Pinkert & Jetz (2022). Some species reported by Sandall, Pinkert & Jetz (2022) are occasional, and should not be considered part of the fauna of a particular region. Other species are clearly wrong records. One example is Ischnura senegalensis, an Afro-Asiatic species that Sandall, Pinkert & Jetz (2022) included in the list for Belgium and The Netherlands, likely due to the individual emerging in an aquarium reported by Adriaens & De Knijf (2020), or Heliocypha vantoli, an Asiatic species included in the species list for The Netherlands (it was not possible to track the origin of that clearly wrong record). Also, Ischnura saharensis is reported for Portugal, but that species has never been found in mainland Portugal or the Macaronesian islands of Azores and Madeira (Weihrauch et al., 2016). Given that the list of Sandall, Pinkert & Jetz (2022) includes all odonate species of the world, it is likely that some errors still remain in their database. However, these minor inconsistences in the number of species of each country are unlikely to affect the overall latitudinal patterns analysed here. References not included in the review of Sandall, Pinkert & Jetz (2022) were maintained because are based on expert knowledge, and additional references published after 2015 were also checked and included.

Given the strong effect that area has on the number of species of a territory (MacArthur & Wilson, 1967), very large countries, over 1,900,000 km2 (Canada, United States of America, Brazil, Argentina, China, India, Indonesia, Russian Federation and Australia) were analyzed by regions/states. In some cases, this was not possible due to the absence of detailed information to partition them (e.g., Algeria, Congo, Kazakhstan, Mexico, Saudi Arabia). The final dataset is available as an excel file in File S1. For each territory the area, latitude and maximum elevation (this last variable only for islands) were compiled using Wikipedia.org (using the coordinates of the capital city when no latitude was included in the description of a territory). Island lists with 10 species or less or with very low counts for their area were closely analysed and discarded if sampling was clearly preliminary (e.g., based on one or two occasional visits). This quality check indicated that for some territories/islands the data were clearly incomplete (N = 10) or anomalous (N = 5) and therefore these areas were excluded from the analysis (see File S1). One island connected to the continent with a sand barrier was also excluded, as the number of species recorded was extraordinarily high (77) for its size of 42 km2, and was considered an outlier (Anjos-Santos & Costa, 2006). Further six Indonesian tiny islands (area 0.03–5.3 km2), situated very close to larger islands (0.5–4.8 km), were also excluded as they were sampled only once, had more than 10 species/km2, even if some had no permanent water (Alfarisyi, 2019), and the author suggested that specimens were vagrants from the larger islands.

To test the predictions of the Theory of Island Biogeography, the distance of islands to the nearest mainland was extracted from Wikipedia.org, or Britannica.com. Distances for most Caribbean islands were obtained from Cineas & Dolédec (2022). For islands not found in the above sources, the distance to the nearest continent (for large islands of archipelagos), or the nearest large island (for small islands, which can be assumed to be the source of colonization) was estimated using Google Earth. This is particularly relevant for complex archipelagos, like the Philippines, Indonesian islands, Fiji, etc. Very large islands, those with an area over 400,000 km2, like Greenland, Madagascar, New Guinea and Sumatra were considered continents for the analysis.

To analyze latitudinal patterns and area effects, the number of species against latitude and area was plotted. A linear model with normal errors was constructed, with the number of Odonata (log x + 1) as the response variable, because some sites had zero species. Predictors were Latitude in absolute value and the logarithm of Area, with Island (yes/no) as a factor. For islands, a similar linear model was constructed including Latitude, Log Area, Distance and Elevation. The distribution of residuals was normal, indicating that the model fitted the data. It has been suggested not to transform count data and analyze them using generalized linear models (GLM) with Poisson error structure instead (O’Hara & Kotze, 2010). However, for this dataset where the number of species varies between zero and several hundred, using a logarithmic transformation is preferable, as it allows to linearize the effect of area on species richness. A GLM with Poisson errors produces similar conclusions in any case (data not shown). To study how the proportion of Zygoptera is affected by the explanatory variables, a GLM with binomial errors was built with the number of Zygoptera species for each territory as the response variable, using the number of Odonata species as binomial totals. As predictors the same variables as above were included. Statistical analyses were done with JASP 0.19.3 (JASP Team, 2024), xlStat 2021 (http://www.xlstat.com) and Genstat 24th edition (GenStat, 2024).

Results

Information was found for 255 countries, states or regions in continents, and 243 islands and archipelagos. No species of odonate has been recorded in Greenland (considered here a continent), neither in five island territories: Bouvet (Norway), Svalbard and Jan Mayen Islands (Norway), Falkland (UK), South Georgia and the South Sandwich Islands (UK) and Heard Island and McDonald (Australia), all situated at latitudes over 51 in absolute value.

As expected, there is a clear effect of latitude, area and insularity on the species richness of Odonates, with a maximum of about 550 species in the equator for continents but only around 200 species in islands (Figs. 2–4; Table 1). The continental areas with highest richness are Venezuela (548) and Colombia (543 species). Brazil (863) and China (818) have higher richness, but given their size were included in the analysis as states and regions. Japan (209) is the archipelago with highest richness, albeit Indonesia (737) and the Philippines (306) have more species, but were included subdivided by islands. New Guinea (470), Borneo (371) and Sumatra (264) have also high richness, but given their size over 400,000 km2 were considered continents for these analyses.

Figure 2 The number of species of Odonates in continents and islands, in relation to latitude.

N = 255 continental areas and 243 islands. The dotted vertical lines represent the tropics of Capricorn (−23.46°) and Cancer (23.46°).

Figure 3 The relationship between area and number of species of odonates for continental areas and islands.

Some outliers, due to very cold climate, are indicated.

Figure 4 Marginal effect of latitude (A), area (B) and insularity (C) on species richness of Odonata (on the original scale).

Table 1 Analysis of the effect of latitude, area and insularity on the number of Odonata species (as log x+1).

Coefficients	
Model	Unstandardized	Standard Error	Standardizeda	t	p	
M0	(Intercept)	3.702	0.060		61.277	<.001	
M1	(Intercept)	2.866	0.200		14.318	<.001	
 	Latitude_abs	−0.024	0.002	−0.291	−10.611	<.001	
 	Log Area	0.206	0.015	0.565	13.449	<.001	
 	Island (yes)	−0.926	0.115	−8.043	<.001	
M2	(Intercept)	4.285	0.385	11.133	<.001	
 	Latitude_abs	−0.019	0.006	−0.228	−2.920	0.004	
 	Log Area	0.093	0.033	0.255	2.838	0.005	
 	Island (yes)	−2.802	0.358	−7.831	<.001	
 	Log Area ∗ Island (yes)	0.183	0.033		5.522	<.001	
 	Latitude_abs ∗ Log Area	−6.068 × 10−4	6.027 × 10−4	−0.095	−1.007	0.314	
Notes.

a Standardized coefficients can only be computed for continuous predictors.

Figure 2 does not consider the effect of area on species richness, except for the fact that very large countries were analysed by regions. The relation between the number of species and area for continental regions and islands is plotted in Fig. 3. Table 1 shows the results of the analysis of the Odonata species richness (Log x+1) against latitude, log area and insularity. The number of species clearly decreases with latitude (Fig. 4A) and insularity (Fig. 4C) and increases with area (Fig. 4B). The model has an adjusted R2 = 0.667, and an ΔAkaike’s Information Criterion (AIC) = 544.32 (difference in d.f. 3) compared to the null model including only the intercept (M0 in Table 1). Plots in Fig. 4 show the effects estimated using the untransformed response variable, for easier interpretation. For the same area and latitude, the insularity effect reduces species richness by around 61% (Fig. 4C). Table 1 also shows the results of a model including the interaction between area and insularity and between latitude and area (model M2). This model is more supported by the data, with an ΔAIC = 30.86 (difference in d.f. 2), but only increases R2 by 2.2%. There is a significant interaction between area and insularity (Table 1) indicating that the slopes for the relationship between species richness and area are different, with steeper increase for islands (Fig. 3). Some clear outliers are Nunavut (Canada), with only two species recorded, and Iceland, with one species (Anax ephippiger, not reproducing in the island; Suhling et al., 2015). Removing these two outliers does not change the conclusions of the analysis (data not shown). The interaction between latitude and area is not significant (Table 1).

Islands have lower species richness (Figs. 2 and 3), and this effect is expected to be more intense for oceanic islands situated at a large distance from continents. A linear model using species richness (log x+1) as the response variable examined the predictive power of latitude, log area, distance from continents (or large islands) and elevation. Results indicate that all factors have a significant effect on species richness, except for elevation (Table 2). This model has an adjusted R2 = 0.613, and ΔAIC=228.45 (difference in d.f. 4) compared to the null model including only the intercept (M0). The marginal effects of area, distance and elevation are plotted in Fig. 5. A further model including the interactions between latitude and area, area and distance, and area and elevation has an adjusted R2 = 0.623, with an ΔAIC = 0.56 (difference in d.f. 3), increasing R2 by 1.0%, and is therefore not supported by the data.

Table 2 Analysis of the effect of latitude, area, distance to continents and elevation on the number of Odonata species in islands.

Coefficients	
Model	Unstandardized	Standard Error	Standardized	t	p	
M0	(Intercept)	2.762	0.071		38.927	<.001	
M1	(Intercept)	2.282	0.120		19.023	<.001	
 	Latitude_abs	−0.029	0.003	−0.381	−9.357	<.001	
 	Log Area	0.208	0.023	0.539	9.156	<.001	
 	Distance	−5.168 × 10−4	5.507 × 10−5	−0.378	−9.385	<.001	
 	Elevation (m)	7.824 × 10−5	6.731 × 10−5	0.068	1.162	0.246	
M2	(Intercept)	2.270	0.166		13.674	<.001	
 	Latitude_abs	−0.019	0.007	−0.252	−2.630	0.009	
 	Log Area	0.214	0.029	0.555	7.313	<.001	
 	Distance	−4.983 × 10−4	1.435 × 10−4	−0.364	−3.473	<.001	
 	Elevation (m)	−3.344 × 10−4	1.833 × 10−4	−0.290	−1.824	0.069	
 	Latitude_abs ∗ Log Area	−0.001	9.659 × 10−4	−0.157	−1.406	0.161	
 	Log Area ∗ Distance	−1.193 × 10−6	2.442 × 10−5	−0.005	−0.049	0.961	
 	Log Area ∗ Elevation (m)	4.457 × 10−5	1.858 × 10−5	0.406	2.399	0.017	

Figure 5 Marginal effect of area (A), distance to continent (B) and maximum elevation (C) on species richness of Odonata in islands.

For elevation the plot shows the (not significant) effect on the logarithm of species richness, as the effect of that factor is significantly positive when the number of species is analysed on the original scale.

Eight out of 10 oceanic islands situated at over 1,500 km from a continent have 0–4 species of Odonata. The exceptions are French Polynesia with 22 species and Hawaii with 37. The most remote islands like Pitcairn (UK) at 5,500 km from the continent and Easter Island (Chile) at 3,600 km have only one odonate, the appropriately named “globe skimmer dragonfly”, Pantala flavescens (Fig. 1C).

The proportion of Zygoptera over the total of Odonata at the different regions, analyzed with a GLM with binomial errors, was negatively affected by latitude (estimate: −0.007, standard error of mean (SE): 0.001, t487 = −6.94, p < 0.001), and positively by the area (estimate: 0.044, SE: 0.010, t487 = 4.48, p < 0.001), but not by insularity (estimate: 0.006, SE: 0.061, t487 = 0.10, p = 0.919). In contrast, in islands the proportion of Zygoptera was not affected by latitude (estimate: −0.002, SE: 0.003, t232 = −0.71, p = 0.478), distance (estimate: 0.0000325, SE: 0.0000717, t232 = 0.45, p = 0.651) or elevation (estimate: −0.0000597, SE: 0.0000509, t232 = −1.17, p = 0.242), but was positively affected by area (estimate: 0.053, SE: 0.024, t232 = 2.27, p = 0.024).

Discussion

The main patterns expected from island biogeography theory are clearly shown by odonates, with the maximum species richness in the equator and a drastic reduction (61%) by insularity (Fig. 4C). Odonates are considered insects of tropical origin (Lorenzo-Carballa & Cordero-Rivera, 2014) and the highest species richness is found in the Oriental and Neotropical biogeographical regions (Kalkman et al., 2008; Suhling et al., 2015). Precipitation and temperature are the main drivers of odonate regional richness, and their combination regulates seasonal and life history evolution of odonates (Beatty et al., 2023).

Results indicate that, as expected, species richness is maximum at the equator, but as Fig. 2 shows, there are many continental areas with lower number of species than expected for a tropical region. Considering only continental areas at latitudes lower or equal to 15° in absolute value, the average number of species is 199, but five regions have less than 50 species, namely Djibouti (10), Burundi (23) and Niger (30) (Sandall, Pinkert & Jetz, 2022), Sergipe (Brazil, 34; Santos et al., 2020) and Paraíba (Brazil, 49; Koroiva et al., 2021). These areas are poor in species are likely due to their low precipitation and/or small size, but in some cases, inventories might be incomplete. In contrast, the average number of species in islands of the same latitudinal range is only 31, with a variation between 1 and 162 species (Java) (Dow et al., 2024), illustrating the effect of insularity.

According to island biogeography, species richness increases with island size, and decreases with distance to the continent (MacArthur & Wilson, 1967; Whittaker, 1998), patterns that have repeatedly been confirmed in many studies (Willig, Kaufman & Stevens, 2003). However other factors, like dispersal ability, are also relevant in this context. For instance, among aquatic insects, Odonata and Coleoptera have larger distribution patterns than Trichoptera and Ephemeroptera (Cineas & Dolédec, 2022), orders with low dispersal ability. Island type is also an important factor. In fact, older oceanic islands have high speciation rates whereas islands once connected to a continent will lose species after insularization (Gillespie & Roderick, 2002). Finally, the effect of island age is illustrated by the younger age of species found on young islands (Heaney, 2007).

“Land islands”, i.e., areas of pristine habitat surrounded by humanized landscapes, may also function as islands in the long term (but see Janzen, 1983), a fact that has many consequences for reserve design and management (Shafer, 1990). Comparing land reserves with oceanic islands might be appropriate for animals with high dispersal ability like Odonates. Samways, Pryke & Simaika (2011) found that, although land islands have more species than oceanic islands for the same size, except for very large islands that act as continents (as is clear in Fig. 3), threats to dragonflies are similar in South African land islands and the islands of the Western Indian Ocean. In both cases, species richness increases with island size, but surprisingly land islands have more range-restricted species, and have a similar proportion of threatened species, which even are subjected to a higher level of threat than oceanic islands of similar size.

Previous studies have analyzed Odonata of some archipelagos and found a clear effect of isolation (Couteyen & Papazian, 2012), geology, sea level changes (Jordan et al., 2005) and endosymbionts (Lorenzo-Carballa et al., 2019) on their phylogeography and species richness. More than 300 species of freshwater invertebrates have been found in the smallest islands of the Caribe, with larger islands having more species, but also a clear effect of island elevation, because streams are only found in islands able to intercept clouds, creating more freshwater habitats (Bass, 2003). However, the present analysis indicates that, for odonates, the effect of elevation is not significant (Table 2). In the Caribbean, Odonata endemism rate ranged between 1.8% (Jamaica) to 42.4% (Tobago) (Cineas & Dolédec, 2022). In the islands around Madagascar, odonate endemism rate ranges from 19% (Seychelles) to 34% (Mascarenes) (Couteyen & Papazian, 2012). Insular radiations of odonates, with a hight proportion of endemism, are also known in very isolated archipelagos, like Hawaii (Jordan, Simon & Polhemus, 2003; Jordan et al., 2005), Vanuatu (Ferguson et al., 2023) and Fiji (Beatty et al., 2017; Donnelly & Marinov, 2024), but even small islands close to continents may have endemic odonates (e.g., Dijkstra, Clausnitzer & Martens, 2007).

This analysis allowed to detect some outliers, which were excluded from the database. Lengkuas island in Indonesia has the record with 13 species of odonates in 0.03 km2 (Alfarisyi, 2019). This tiny island is only 4.8 km from Belitung Island, and all species found were common libellulids, with one exception (Libellago hyalina). Given that the island has no permanent freshwater (Alfarisyi, 2019), these are likely vagrant individuals. This example highlights the fact that odonates can be found on small islands even if they have no suitable habitats to reproduce, and are able to colonize artificial containers very fast (Jackson, 1968). A more surprising case is the island of Marambaia in Brazil, that has 77 species of odonates in only 42 km2 (Anjos-Santos & Costa, 2006). This extraordinary diversity might be explained by its environmental complexity, with many wetlands, an elevation of 641 m allowing the formation of rivers and the fact that it is connected to the mainland by a sand barrier and is very close to a highly diverse area. While wide sea channels, very large rivers (Beatty et al., 2023) or simply open areas are certainly a barrier to the dispersal of some species, like the forest species (Figs. 1B, 1D), or the giant Pseudostigmatidae which have low flying endurance (Fincke, 2006), narrow sea channels do not impede odonate dispersal.

The ratio of Zygoptera (Figs. 1A, 1B) to Anisoptera (Figs. 1C, 1D) is well stablished as an indicator of community disturbance, at least in Tropical America (Ribeiro, Juen & Rodrigues, 2021; Machado De Albuquerque et al., 2024; Pires et al., 2025), due to the fact that many damselflies are forest specialists (Fig. 1B), which rarely disperse to open areas because of thermal requirements (De Marco Júnior, Batista & Cabette, 2015). This ratio was negatively affected by latitude and positively by the area, but not by insularity. This means that Anisoptera tend to dominate in areas far from the equator, likely because they are bigger (and some endothermic; May, 1979), but damselflies are more common in larger regions. The proportion of damselflies to dragonflies was similar in continental areas and islands, suggesting that, the dispersal ability (Góral, 2024) (and/or the speciation ratio) of both suborders is similar. In contrast, in islands the proportion of Zygoptera was not affected by latitude, distance or elevation, but was positively affected by the area, as in continents. Very large islands normally have a higher proportion of damselflies, like Philippines (1.71 damselflies/dragonflies), New Guinea (1.64), Indonesia (1.45) and Madagascar (1.25), but some exceptions are Sumatra (0.74), Java (0.54) or Japan (0.41).

Despite the clear effect of insularity on species richness, Odonates are well known as island colonizers (Samways, 2003), able to establish in new habitats in a couple of years (Samways, 1998). Ten species of odonates (26 specimens, four species of damselflies) were collected over the Andaman sea during a tropical storm, when they perched on a ship (Ruddek, 1998), indicating that these species can be dispersed by storms over large distances. An American damselfly, the only parthenogenetic species of odonate known, Ischnura hastata (Fig. 1A) has been able to colonize the Azores archipelago (Cordero-Rivera et al., 2005) and also the Galapagos, where it has sexual populations (Cordero-Rivera et al., 2023), illustrating the ability of this species to disperse over the oceans. In fact, a total of 21 specimens of odonates were captured on nets mounted in airplanes, including six individuals of I. hastata, three of them at 304 m of altitude, and one unidentified zygopteran at 914 m (Glick, 1939). These facts and the presence of damselflies (particularly Ischnura, Cordero-Rivera, 2015) in many oceanic islands, clearly indicate that the dispersal abilities of some damselflies are underestimated (Góral, 2024). However, the species most frequently found in the most isolated archipelagos is a dragonfly, Pantala flavescens (Fig. 1C), which was recorded in the remote Amsterdam island, southern Indian Ocean (Devaud & Lebouvier, 2019), which is the only odonate present in Easter island (Samways & Osborn, 1998), and is a regular long distance migrant (Hobson et al., 2021). However, the records of P. flavescens and Pantala hymenaea for the islands of Saint Pierre et Miquelon (Sandall, Pinkert & Jetz, 2022), situated at a latitude of 46.9°N, are likely to be wrong. Sandall, Pinkert & Jetz (2022) also inaccurately include three tropical species of diverse biogeographical regions (Amphicnemis valentitni, Euphaea cyanopogon and Heteragrion bickorum) in their database for Iceland, and even more surprisingly include Argia carolus, Argia elongata and Protoneura cara for Saint Helena, Ascension and Tristan da Cunha. This last fact is specially astonishing, because only one odonate has been recorded for Saint Helena, Sympetrum dilatatum, the only odonate considered extinct by IUCN, which was endemic to the island, and was apparently extirpated by the introduction of an invasive exotic frog (Pryce, 2021). Furthermore, Diceratobasis macrogaster, an endemic species of Jamaica has been erroneously included in the dataset for Cuba (Cineas & Dolédec, 2022). Therefore, some caution is needed when using these databases, to use only data that are well supported by experts.

Conclusions

In conclusion, this review of the species lists for continental regions and islands confirms that latitudinal patterns of species richness are clearly shown by Odonates, a group of insects with one of the highest dispersal abilities (Kalkman et al., 2008). However, the effect size of latitude is small (Table 1) and there is high variation among areas (Fig. 2) that needs to be studied in more detail. Islands show reduced species richness (by an average of 61%), but also a high degree of endemism, particularly in very isolated archipelagos. Both suborders show similar patterns in relation to island biogeography, as in both cases there are families with high ability to colonize oceanic islands (Libellulidae and Coenagrionidae). However, dragonflies (Anisoptera) are better colonizing cold regions, and damselflies (Zygoptera) are more speciose in countries with higher land area.

Supplemental Information

Supplemental Information 1 Dataset of number of species of Odonates by region

I would like to thank Oleg Kosterin for providing a map of the ecoregions of Russia, used to estimate their area to include these regions in my analyses, and to Milen Marinov for unpublished information about the species richness of Fiji.

Additional Information and Declarations

Competing Interests

Author Contributions

Data Availability

The authors declare there are no competing interests.

Adolfo Cordero-Rivera conceived and designed the experiments, performed the experiments, analyzed the data, prepared figures and/or tables, authored or reviewed drafts of the article, funding, and approved the final draft.

The following information was supplied regarding data availability:

The raw data are available in the Supplementary File.

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
