# Peer review of "Global species richness of dragonflies and damselflies (Odonata): latitudinal trends and insular colonization"

_PeerJ, doi:10.7717/peerj.20004_

## Round 0.1 · original submission · Major Revisions

There are a lot of comments by the reviewers please have look at these.

·

Basic reporting

Clear and unambiguous, professional English used throughout.
Line 25: I recommend adding non-insular before “regions”, to clearly distinguish the results stated here with the insular results fond in lines 26 to 28.
Line 32: The spelling of “stablished” should be changed to established.
Line 35: I recommend changing “large” by larger.
Lines 42-43: I recommend ending the sentence after “(Terborgh, 1992)”, and starting the next sentence with In fact, the effect of… to ease the reading of the paragraph.
Line 244: There could be commas missing before and after “for odonates”.
Line 308: I recommend changing the spelling of “Saint Pierre and Michelon” to the common spelling of Saint Pierre et Miquelon.

Literature references, sufficient field background/context provided.
Lines 119-121: The author partitioned large countries into regions, when possible, to avoid species-area effect, which I second. However, while the species-area effect is a basic biogeography tenet, I recommend adding a citation specific for dragonflies to justify this statement.

Professional article structure, figures, tables. Raw data shared.
Figure 2 shows the relationship between the number of species and the latitude of the checklist’s site. The results of this study show a clear linear relationship between absolute latitude and number of species. Therefore, I recommend changing the x-axis unit to absolute latitude, rather than latitude, as it would make it easier to visualize the statistically significant results.
Figure 3 shows the relationship between log area and log number of species + 1, grouped by continents of islands. I recommend adding the confidence intervals for both curves and change the equations of each group by the R2 values, which could be more visually useful.

Certain changes to the supplementary materials would make the information completer and more useful to use in further studies.
• Adding a column with the longitude would increase the usefulness of this data for further studies.
• Due to the phenological patterns that odonates exhibits, especially in temperate regions, I recommend adding, when possible, a column specifying the season or month when the checklist was made.
• Row 529 of the supplementary materials specifies “sites excluded for different reasons”. I recommend adding a column specifying the reasons why each site was excluded.
• Adding a metadata sheet to explain the meaning of each column would help ease the interpretation of the data frame. For example, it is unclear what the Continent-yes and Island-yes columns inform of.
• I recommend adding missing units to the columns Continent-yes, Island-yes, Distance and Log_Area.

Self-contained with relevant results to hypotheses.
Due to the descriptive nature of this study, I believe that specifying predictions and hypotheses is not mandatory.

Experimental design

Original primary research within Aims and Scope of the journal.
No comment.

Research question well defined, relevant & meaningful. It is stated how research fills an identified knowledge gap.
No comment.

Rigorous investigation performed to a high technical & ethical standard.
No comment.

Methods described with sufficient detail & information to replicate.
Lines 119-121: I recommend adding a minimum area threshold for the partition of large countries into areas and regions.

Lines 149-151: The author includes sites with odonate checklists of zero species. However, there are many islands across the globe where dragonflies are unlikely to establish permanent populations due to niche availability, such as low temperatures or lack of freshwater. I recommend the exclusion of sites with 0 species or adding a justification for the inclusion of certain sites with 0 species.

Lines 156-159: Six islands were excluded from this study for being too small (0.03-5.3 km2), too close to larger islands (0.5-4.8 km), for having too many species per square kilometer (>10 species/km2), and for lacking permanent water. However, in the supplementary information, I found 14 islands smaller than 5.3 km2, 15 islands closer than 4.8 km, and 2 islands with more than 10 species / km2. I recommend explaining in more detail why these 6 islands were excluded or explaining why the aforementioned islands were included.

Line 166-167: Islands greater than 400,000 km2 are considered as continents. While I understand that up to certain threshold any landmass can be considered an island, I recommend adding a justification for this threshold.

Lines 256-257: The author excluded some islands lacking freshwater with vagrant individuals, which I second in the context of this study. However, in the supplementary materials, I identified the study Devaud, M., & Lebouvier, M. (2019). First record of Pantala flavescens (Anisoptera: Libellulidae) from the remote Amsterdam Island, southern Indian Ocean. Polar Biology, 42(5), 1041–1046. https://doi.org/10.1007/s00300-019-02479-3. This study states that the individuals found are vagrant and likely would not be able to establish a permanent population due to low temperatures. I recommend omitting this checklist for the purposes of this study to keep outlier-deletion consistent, and double-checking studies included in this meta-analysis to further exclude studies that should be omitted based on the criteria of the author.

Validity of the findings

Impact and novelty not assessed. Meaningful replication encouraged where rationale & benefit to literature is clearly stated.
No comment.

All underlying data have been provided; they are robust, statistically sound, & controlled.
Model selection for the GLMs is not specified. In lines 180 and 189, ΔAIC values are provided, but in the methods section, it is not specified that model selection uses ΔAIC. I recommend adding the method of model selection in the methods sections. Additionally, while the author specifies what response variables are significative, they do not test if there is an interaction or not between them. Understanding the interaction between response variables is crucial for the interpretability of the results, therefore I would strongly recommend adding these statistical results.

I strongly recommend running the diagnostic analyses on the residuals of the generalized linear models chosen, to ensure that the model chosen fit the assumptions of GLMs, and that the log transformations + 1 are adequate for the raw data collected.

Line 160: The author states an effect of latitude and insularity on species richness but does not provide the statistical results to back this statement. I recommend adding these statistical results.

Conclusions are well stated, linked to original research question & limited to supporting results.
By addressing the statistical concerns mentioned, I believe the conclusions will be more strongly supported.

·

Basic reporting

A great paper using existing collections data to ask a new question. It is also an excellent example of asking what seems like a very common questions, except on a new taxonomic group (i.e., Odonata).
The English language, especially punctuation and grammar, should be improved to ensure that an international audience can clearly understand your text. Some examples where the language could be improved include lines 83-86, 116, 122, 229-230, 277, 300, – the current phrasing makes comprehension difficult.
The use of references in the introduction do a fine job of illustrating how the current question fits within current theory and practice.
The hypothesis is clear and the results and conclusion are related to it.

Experimental design

The experimental design is clearly explained in the Materials and Methods section.
The statistical methods and parameters are clearly delineated.
Some relation to how these methods match other latitudinal diversity studies is necessary. Is the 400,000 square kilometer definition of a continent standard?
Some more explanation of Zygoptera versus Anisoptera questions and results is necessary. Perhaps including multiple versions of the figures would help. What do Figures 2, 3, and 4 look like when the data is divided among Zygoptera and Anisoptera? The results text does not make clear how the patterns differ between these two groups.

Validity of the findings

More details on how these results compare to similar studies in other groups is necessary. For example, at line 226 "other studies" are mentioned, but how do the present linear models compare to those other studies. Are Odonata exactly the same as other insects or completely different? Is it different because it is aquatic? Is it the same as other aquatic groups?
The discussion and conclusion are based on the results. The initial hypothesis is addressed. A clear simple answer is given, but it still must be linked how it is the same or different from other insect (or non-insect groups).

Additional comments

Thank you for giving me the chance to review this paper. I am always encouraged to see researchers making valuable use of the masses of specimen data that exist in collections and online databases.

Reviewer 3 ·

Basic reporting

The manuscript aims to investigate the latitudinal gradient of diversity, effects of insularity, and the Theory of Island Biogeography in odonates at a global scale.
Overall, the manuscript provides extensive background information and context for the study’s questions. However, at times this information becomes excessive and takes away from the clarity of the manuscript. Further, there are issues with grammar, sentence structure, and the general logical flow of the paragraphs that interfere with the readability of the manuscript. Some context on the Theory of Island Biogeography is also missing in the introduction. I have provided my comments below:

L32-38: I believe it would benefit the manuscript to provide an explicit explanation of the latitudinal gradient of diversity before delving into how it varies across ecosystems, causes, etc.
L39-40: In general, do insects follow the expectations of the latitudinal gradient of diversity? I think it would be helpful to state if the examples provided are exceptions or tend to be the norm for insects.
L41-58: From my understanding, this section is aiming to provide potential hypotheses for how odonate diversity may vary along a latitudinal gradient. I believe it would improve the clarity of the manuscript to first state the odonates are understudied with regards to the latitudinal gradient of diversity and then describe odonate ecology (e.g., how they are carnivorous and predaceous) and how their ecology may drive odonate diversity along latitudinal gradients. Further, the manuscript discusses how being predaceous in tropical environments will result in more intense biotic interactions, but it is not clear to me how this relates to species richness.
I would also suggest cutting out sentences that are discussing trends in mayflies, caddisflies, and stoneflies since the manuscript’s focus taxa are odonates. If the author would prefer to leave this context, I believe it may improve the readability of the manuscript to condense the information further and make explicit connections to how trends in these aquatic insects relate to potential trends in odonates.
L62-76: It is my opinion that the information in this section is excessive. I would suggest replacing this with context on insularity and island biogeography in odonates as this is currently missing from the introduction.
L84 and L87: 'Scholar Google' should be 'Google Scholar'

Finally, I kindly suggest that the author considers seeking a professional English editing service or a colleague with full professional proficiency in English to review the manuscript.

Experimental design

The methods are clear and described in detail. My comments are below:
L125-127: The manuscript states, “Island lists with 10 species or less or with very low counts for their area were closely analysed and discarded if sampling was clearly preliminary (e.g. based on one or two occasional visits).” Could the author elaborate on what “closely analysed” means? I would suggest describing the full procedure for determining if data should be discarded.
L128-133: I would suggest stating that the purpose of collecting these data is to study the effect of insularity on odonate diversity.
L136-144: Per Kotze & O’Hara, 2010, I would suggest using a generalized linear model instead of log-transforming the species richness data and using a simple linear regression.
L141-144: It’s a bit unclear to me if a ratio of Zygoptera to total odonates is being used here. It is later clarified in the text, but I suggest making it clearer in the methods that this is the case.

Validity of the findings

I believe the conclusions need more nuance, but are, in general, supported by the results. My comments are below:
I would add in conclusions from the analyses on suborder.
L182-183: The manuscript states that insularity reduces species richness by ~64% and cites Figure 4c. Later, the manuscript repeats this statistic and cites Figure 2. It is not totally clear where this percentage is being estimated from, as it is not discernable from Figure 2 or Figure 4c. From Figure 4c, I would approximate that the percent decrease is ~54%, since the ‘no Island’ richness is ~110 and the ‘yes Island’ richness is ~50, and 110-50/110 = 0.545. Could the author clarify this?
L326-328: I agree with the author that the results show that odonates follow latitudinal patterns of diversity; however, the effect size seems a bit weak. I would suggest clarifying the conclusion with this.

Additional comments

L151-159: I would suggest moving exclusion criteria to the methods section.
L162-167: It is my opinion that the manuscript needs to be clearer about how continents and islands are defined. Countries are labeled as continents or islands as they are referenced in the text, but no size criteria are explicitly described in the methods. This would aid in the clarity of the text.
L184-187, L197-198 are describing models that are being used. I suggest reporting only the results from the models as the author has already described the models in the methods.
The discussion section is a bit disorganised. L285-323 provide interesting cases where odonates were able to quickly establish new communities in isolated habitats, but this section is, in my opinion, more detailed than it must be. It would improve the clarity and readability of the manuscript to condense this section.

---

## Round 0.2 · Minor Revisions

Please look at reviewer #3's comments about using the GLM. Can you respond to their specific comments?

·

Basic reporting

I believe the manuscript has been improved, with all my comments taken into consideration, and with changes made when needed. I agree with the author's justifications for when changes I suggested were not made. Overall, it's an excellent paper that informs on fundamental trends in odonate biogeography.

Experimental design

No further comments.

Validity of the findings

No further comments.

Additional comments

No further comments.

Reviewer 3 ·

Basic reporting

I thank the author for addressing my concerns and their work on the manuscript.

Clear and unambiguous English are used throughout the text, there are sufficient context, and the manuscript has a professional article structure.

My only comment would be that L309-349 could be condensed further. I actually thought the dragonfly colonization lore was extremely interesting, but it might be best to make this section more concise in order to wrap up the manuscript neatly.

Experimental design

My only comment is about the choice of model. I’ve read the revision of this section (L158-164) and the author’s response. I still recommend using a GLM with either a Poisson or negative binomial distribution. A GLM would handle the 0s in the data without the use of an arbitrary constant, as well as a non-normal distribution of the species richness measurements. While I understand that adding a constant is a common workaround when log-transforming data with zeroes, doing so makes the interpretation of the model murky. Specifying a log-link within a GLM would address the author’s goal of linearizing the relationship between species richness and the predictors without transforming the data and avoids an arbitrary constant.

Even if the model estimates are similar between a GLM and the log-transformed data, if the distribution of the original species richness data are not normally distributed, I believe that a GLM would be a more robust and grounded framework for the manuscript’s questions. Could the author provide diagnostics of the GLM with a Poisson and negative binomial distribution and compare with diagnostics of the transformed data? I recognize this would be a bit more on the author's plate, so I would be grateful if this is possible.

Validity of the findings

Good comparison of the odonates to other aquatic macroinverts.

I appreciate the edits made to the manuscript to clarify the results (e.g., discussing the variability in the conclusion).

---

## Round 0.3 · accepted · Accept

Thank you for addressing the comments from the reviewer and taking the time to show the two different approaches, and why you are staying with your approach. In my opinion the manuscript is ready for publication.